# Role of NAD^+^—Modulated Mitochondrial Free Radical Generation in Mechanisms of Acute Brain Injury

**DOI:** 10.3390/brainsci10070449

**Published:** 2020-07-14

**Authors:** Nina Klimova, Adam Fearnow, Tibor Kristian

**Affiliations:** 1Veterans Affairs Maryland Health Center System, 10 North Greene Street, Baltimore, MD 21201, USA; nklimova@umaryland.edu (N.K.); afearnow@som.umaryland.edu (A.F.); 2Department of Anesthesiology and the Center for Shock, Trauma, and Anesthesiology Research (S.T.A.R.), University of Maryland School of Medicine, Baltimore, MD 21201, USA

**Keywords:** mitochondria, NAD^+^, free radicals, acetylation, ischemia

## Abstract

It is commonly accepted that mitochondria represent a major source of free radicals following acute brain injury or during the progression of neurodegenerative diseases. The levels of reactive oxygen species (ROS) in cells are determined by two opposing mechanisms—the one that produces free radicals and the cellular antioxidant system that eliminates ROS. Thus, the balance between the rate of ROS production and the efficiency of the cellular detoxification process determines the levels of harmful reactive oxygen species. Consequently, increase in free radical levels can be a result of higher rates of ROS production or due to the inhibition of the enzymes that participate in the antioxidant mechanisms. The enzymes’ activity can be modulated by post-translational modifications that are commonly altered under pathologic conditions. In this review we will discuss the mechanisms of mitochondrial free radical production following ischemic insult, mechanisms that protect mitochondria against free radical damage, and the impact of post-ischemic nicotinamide adenine mononucleotide (NAD^+^) catabolism on mitochondrial protein acetylation that affects ROS generation and mitochondrial dynamics. We propose a mechanism of mitochondrial free radical generation due to a compromised mitochondrial antioxidant system caused by intra-mitochondrial NAD^+^ depletion. Finally, the interplay between different mechanisms of mitochondrial ROS generation and potential therapeutic approaches are reviewed.

## 1. Introduction

Mitochondria are essential organelles in determining continuous cell survival and cell death. Conditions of many diseases are linked to mitochondrial dysfunction and mitochondrial abnormalities. Therefore, by improving mitochondrial functions and physiology one can significantly mitigate the disease pathology. Mitochondria are critical for several essential cellular processes, including intracellular metabolic activities, and signal transduction of several cellular pathways. They are also involved in cellular ion homeostasis, oxidative stress, and both apoptotic and necrotic cell death. Therefore, disturbance in any mitochondrial function can have detrimental consequences for cell survival.

One of the generally accepted adverse effects of mitochondrial activity under disease conditions is the increased mitochondrial free radical production. An excessively high level of reactive oxygen species (ROS) is commonly accepted as a significant contributor to cell death mechanisms in acute brain injury or in neurodegenerative diseases. Since ROS are generated also under normal physiological conditions, mitochondria possess an ability to eliminate these toxic molecules. However, it should be stressed that free radicals also serve as signaling molecules regulating vital biochemical processes, cellular growth, differentiation, and elimination of damaged cells [1]. Consequently, for normal cellular and mitochondrial functions the level of free radicals is fine-tuned by the two opposite processes, the rate of ROS generation, and the effectiveness of their enzymatic elimination.

There are several mechanisms by which mitochondria can generate an excess of free radicals, and the individual mechanisms can have different contributions to the increase in ROS production under different disease conditions. In this review we will briefly examine the mitochondrial sources of ROS, and the basic mechanisms that constitute the mitochondrial antioxidant system. Furthermore, the role of NAD^+^ and acetyl-CoA metabolism in modulation of post-translational modifications of enzymes participating in the mitochondrial defense system against free radicals will be discussed.

Finally, we will outline the potential therapeutic implications and new targets that can mitigate the adverse effects caused by free radicals due to changes in NAD^+^ metabolism.

## 2. Mitochondrial Oxidative Phosphorylation and ROS Production

Mitochondria are the primary organelles for the generation of adenosine triphosphate (ATP) under normal aerobic conditions. They possess the terminal oxidative pathway for carbohydrate and β-oxidation of fatty acids that produce the reducing equivalents NADH and flavin adenine dinucleotide (FADH_2_). During oxidative phosphorylation (OXPHOS) the energy stored in the reduced compounds is used by the electron transport chain to pump hydrogen ions across the inner membrane, generating the electrochemical gradient (Figure 1). At complex IV the electrons are transferred to O_2_, which is reduced to H_2_O (Figure 1). The backflow of H^+^ into the mitochondrial matrix down their electrochemical gradient through ATP synthase produces ATP (Figure 1).

To regenerate the mitochondrial membrane potential, NAD^+^ and FAD are converted by tricarboxylic tricarboxylic acid (TCA) cycle enzymes back to their reduced forms. Under physiological conditions, low levels of ROS are formed by the leak of electrons from the respiratory chain enzymes to oxygen molecules, generating superoxide radicals (O_2_^•-^) via univalent reduction of O_2_ [2,3].

Apart from the respiratory complexes, additional sources of superoxide are TCA cycle enzymes α-ketoglutarate dehydrogenase (αKGDH), pyruvate dehydrogenase (PDH), aconitase, and glycerol-3-phosphate dehydrogenase (GPDH) [4,5,6,7]. It should be noted that dihydrolipoamide dehydrogenase (DLD) as a component of PDH, αKGDH, and branched-chain αketoacid dehydrogenase complex is a common source of ROS [8]. Other mitochondrial components that are likely to contribute to ROS generation are monoamine oxidase (MAO) and cytochrome b5 reductase in the outer mitochondrial membrane [9], nicotinamide adenine dinucleotide phosphate (NADPH) oxidase 4 (Nox4), and dihydroorotate dehydrogenase [10]. However, the major contributors to mitochondrial ROS are the respiratory chain complexes; much lower contributions come from the above-listed enzymes [11]. Superoxide is generated mainly in complexes I and III [12,13,14,15]. However, significant production of ROS by complex III was observed mainly under artificial conditions when using complex III inhibitor antimycin A [6]. Furthermore, under specific conditions of low succinate concentration and a downstream block of the electron flow in the respiratory chain, complex II can also produce high levels of ROS [16,17,18]. There are several other highly reactive oxidants produced in the mitochondria that include reactive nitrogen species, carbonate radicals, and singlet oxygen (for a review, see [19]). Since the superoxide anions are the major form of ROS generated by mitochondria, we will focus mainly on their generation and detoxification mechanisms.

## 3. Mitochondrial Antioxidant Mechanisms

Under physiological conditions, a net amount of superoxide produced by mitochondria is determined by the rate of O_2_^•-^ generation and the rate of O_2_^•-^ detoxification. Since ROS are involved in normal cellular functions via the role as signaling molecules in cellular protection [20,21], moderate fluctuations in ROS levels are essential for cell physiological responses and survival. To maintain the superoxide levels within physiological concentrations, there is an endogenous antioxidant defense mechanism that scavenges superoxide [4,22,23]. This antioxidant mechanisms in brain mitochondria include the matrix manganese superoxide dismutase (MnSOD), glutathione (GSH), glutathione reductase (GR), glutathione peroxidase (GPX) [24], and the thioredoxin system (Figure 2) [25]. MnSOD plays an essential role in protecting against oxidative stress by converting superoxide to hydrogen peroxide (H_2_O_2_). In the next step H_2_O_2_ is converted to H_2_O by the GPX using GSH as the reducing equivalent or by thioredoxin peroxidase [26]. The oxidized glutathione (GSSG) is then converted back to its reduced form (GSH) by GR using NADPH. This depends on a reducing environment maintained by a high NADPH/NADP ratio [27]. The generation of NADPH from NADP^+^ is driven by NAD(P)^+^ transhydrogenase activity (Figure 2). However, NADP-malic enzyme and NADP-dependent isocitrate dehydrogenase (IDH2) also converts NADP^+^ to NADPH [28]. Interestingly, both the superoxide generation, particularly by respiratory complexes, and superoxide scavenging systems are dependent on a NADH/NAD^+^ ratio [29]. Thus, both the mitochondrial ROS production and mitochondrial free radical detoxification are significantly affected by changes in mitochondrial NAD^+^ levels and its redox state.

## 4. Effect of Ischemia on Mitochondrial Metabolism

Under conditions of global cerebral ischemia, induced by cardiac arrest, the brain tissue oxygen is depleted within a couple of seconds [30]. This is because the amounts of glucose and other substrates that support mitochondrial oxidative phosphorylation are significantly higher than available oxygen for their oxidation [31]. Oxygen deficit then strongly affects mitochondrial metabolism. The electron transport between respiratory complexes stops since there is no oxygen to accept the electrons from complex IV. This results in a complete reduction of all components of the respiratory chain and consequently of NAD(P)H. However, the metabolic substrates for mitochondrial bioenergetic metabolism are still available. Therefore, ultimately there is an accumulation of upstream intermediates such as lactate, succinate, free fatty acids, acyl-CoA, and glycerol [32,33,34,35,36]. This ischemia-altered metabolic stage then affects the mitochondrial respiratory activity and free radical generation during the early reperfusion period [36].

### 4.1. Two Phases of Post-Ischemic Mitochondrial Respiratory Failure

Since mitochondria provide the majority of intracellular ATP via oxidative phosphorylation, the cellular ATP levels rapidly fall after onset of global cerebral ischemia [35,37,38]. The lack of ATP results in arrest of energy-requiring ion pumps and collapse of ion gradients across the plasma membrane. As a consequence, brain cells, particularly neurons, are overloaded with calcium, triggering excitotoxic cell death mechanisms [39,40]. Restoration of oxygen by resuming blood flow is a prerequisite for brain tissue rescue. Re-introduction of oxygen initiates mitochondrial respiration and ATP production. However, it was demonstrated that the pathophysiology of primary intra-ischemic energy failure due to absence of oxygen is aggravated by the recirculation [37]. During reperfusion the brain’s ATP level and mitochondrial respiration do not recover completely [41], and additionally there is a delayed secondary energy failure represented by a decline in ATP levels and a decrease in mitochondrial oxidative phosphorylation [37,42].

Isolated mitochondria from ischemic brain tissue show a significant decline in their respiratory capacity [43,44,45,46,47]. About 30 to 60 min after the start of recirculation there is a normalization of mitochondrial respiratory functions [43,45,48]. However, a secondary deterioration of mitochondrial respiration is observed much later in selectively vulnerable areas [45]. In animals with longer periods of ischemia, recirculation may not restore normal mitochondrial respiration or can result in rapidly developing secondary dysfunction [43,44,47]. Isolated mitochondria then show damage to respiratory complexes and to pyruvate dehydrogenase complex activity [49,50]. Thus, although there is almost full recovery of mitochondrial respiration after 30 min of reperfusion following complete brain ischemia [43], a secondary gradual decrease in mitochondrial respiration is observed several hours after the ischemic insult [45,51]. The inhibition of mitochondrial respiration was observed when supported by substrates of NAD^+^-dependent dehydrogenases that suggests a dysfunction of complex I. This inhibition of complex I activity is one of the causal effects that leads to an increase in post-ischemic superoxide production, particularly during early recovery period (for a review, see [36]).

### 4.2. Mitochondrial Free Radical Production and Ischemic Brain Injury

During early reperfusion, high intramitochondrial succinate levels lead to reverse electron transfer (RET) from complex II towards complex I. This causes a loss of complex I flavin mononucleotide (FMN) [52,53] and a high rate of ROS generation [54,55,56]. Thus, the early stage of reperfusion is associated with a transient rise in ROS levels produced mainly by mitochondria (for a review, see [36]). The FMN-deficient complex I is not able to catalyze physiological NADH oxidation, leading to reduced efficiency of mitochondrial respiration and ATP production.

High levels of ROS can cause peroxidative changes to proteins, lipids, and DNA [57,58,59]. Mitochondrial DNA is one of the main targets for free radicals-induced oxidative damage [60]. This is due to the lack of protection by histones, limited capacity of DNA repair mechanisms, and proximity to the production sites of superoxide. Oxidative stress results in DNA strand breakage and activation of poly-ADP-ribose polymerase-1 (PARP1). This enzyme uses NAD^+^ as a substrate to generate poly-ADP-ribose and activates the DNA repair process [61]. Overactivation of PARP1 depletes cellular NAD^+^ pools and is considered one of the major mechanisms that contributes to ischemic cell death [62,63,64]. Data showing a preservation of post-ischemic NAD^+^ levels in PARP1 knockout animals, and animals treated with PARP1 inhibitors, suggest a significant contribution of this enzyme to post-ischemic NAD^+^ catabolism [62,63,65].

Interestingly, the NAD^+^ levels are reduced also in mitochondria isolated from post-ischemic brain tissue [56,66]. The depleted mitochondrial NAD^+^ pools can reflect the excessive activity of mitochondria localized PARP [67,68,69]. Another mechanism that can contribute to reduction of intramitochondrial NAD^+^ levels is the activation of the mitochondrial permeability transition pore (MPT). The MPT is a large conductance channel formed in the inner mitochondrial membrane that allows solutes of molecular weight up to 1500 Da to diffuse from the mitochondrial matrix into the cytosol [70,71,72]. Conditions during the early reperfusion period promote the MPT activation and consequently loss of mitochondrial NAD^+^ [72,73,74]. Significant reduction of matrix NAD^+^ pools has an inhibitory effect on enzymatic reactions that require NAD^+^ as a cofactor. Thus, the activity of pyruvate dehydrogenase complex (PDHC), respiratory complex I, and enzymes of the TCA cycle will be compromised [75]. Additionally, enzymes that use NAD^+^ as a substrate will be negatively affected.

The opening of the MPT pore leads to mitochondrial depolarization. Since the reverse electron transport is reduced under low membrane potential one would expect that the ROS generation by this process will be inhibited. However, our published data demonstrate that brain mitochondria display a clear-cut heterogeneity in sensitivity to MPT [68]. Therefore, it is feasible that a subpopulation of mitochondria that are less sensitive to MPT will contribute to the rise in ROS due to reverse electron transport.

Generally, the inhibition of the mitochondrial respiration will reduce consumption of the oxygen delivered to the post-ischemic tissue, resulting in increased brain tissue oxygen tension. This rise in brain tissue oxygen levels during reperfusion was observed following transient focal ischemia with up to 60% increases above the physiological values [76]. These hyperoxic conditions then further facilitate the high ROS generation.

### 4.3. Role of Protein Acetylation in Mitochondrial ROS Generation

Lysine acetylation is a post-translational modification that regulates enzyme activity and also expression levels [77,78,79]. This process is determined by two opposing enzymatic activities of histone acetyl transferases (HATs) and histone deacetylases (HDACs). HATs use acetyl-CoA to acetylate lysine of the target protein, while class III HDACs, sirtuins, require NAD^+^ for their deacetylase activity [69,80,81,82]. The Sirt family of proteins is comprised of several members that are localized in different subcellular compartments [82,83]. Sirt3, Sirt4, and Sirt5 were identified as mitochondrial deacetylases [79]. However, only Sirt3 is considered to be the major mitochondrial deacetylase [84,85,86].

Increased acetylation of mitochondrial proteins has an inhibitory effect on their activity [87]. Since the NAD^+^ levels are significantly reduced following ischemic insult, mitochondrial proteins show an excessive acetylation during the first 24 h of recovery [56]. Apart from respiratory complexes and TCA cycle enzymes, enzymes of the mitochondrial antioxidative system, particularly MnSOD, are also hyperacetylated [56]. The increased acetylation of MnSOD inhibits the activity of this enzymes [88], and as a result, mitochondrial capacity to detoxify superoxide is compromised and mitochondria become a hot spot for ROS production (Figure 3) [56,86].

Similarly to the post-ischemic, two-phased mitochondrial respiratory dysfunction, there is an increase in ROS generation during early reperfusion period, and a secondary, several-hours-delayed rise in free radical production following ischemia [56]. Interestingly, the delayed increase in ROS levels coincides with significant reduction of mitochondrial NAD^+^ pools and hyperacetylation of MnSOD [56,65]. The causal link between the delayed, high ROS levels and mitochondrial NAD^+^ depletion was confirmed by treatment of animals with NAD^+^ precursor nicotinamide mononucleotide (NMN) that prevented the post-ischemic reduction of mitochondrial NAD^+^ levels, and inhibited the post-ischemic increase in acetylation of mitochondrial protein and in delayed superoxide production [56,65]. Furthermore, a genetic mouse model with the SIRT3 knockout gene showed hyperacetylation of mitochondrial proteins and high levels of free radicals, and NMN treatment did not have any significant effect on mitochondrial protein acetylation or ROS levels in this mouse model [56]. Thus, by counteracting the post-ischemic mitochondrial NAD^+^ consumption, one can prevent the increase in mitochondrial protein acetylation and maintain the mitochondrial superoxide detoxification efficiency. Additionally, since H_2_O_2_ detoxification depends on NAD(P)H availability, the depletion of mitochondrial NAD^+^ pools after ischemia will impinge the removal of hydrogen peroxide and can lead to an increase in hydroxyl radicals.

Mitochondrial specificity as a source of superoxide was confirmed by using a transgenic mouse model that expresses mitochondria targeted yellow fluorescent protein (mito-eYFP) [89]. These animals were injected with dihydroethidium (DHE) after ischemic insult, and the red fluorescent hydroxyethidium, the reaction product of superoxide and dihydroethidium (DHE) [90], was colocalized with the mitochondrial fluorescent marker [56]. The mitochondrial source of ROS during the delayed recovery period was also confirmed by administration of mitochondria-targeted antioxidant MitoQ that accumulates in mitochondria and scavenges superoxide [56]. Interestingly, hydroxyethidium accumulated specifically in hippocampal pyramidal neurons and interneurons [56], suggesting that the most dramatic increase in superoxide generation was taking place in neuronal mitochondria and that the degradation of NAD^+^ in the post-ischemic brain was probably due to reduction of NAD^+^ pools preferentially in mitochondria localize within neurons.

One of the pathological outcomes caused by increased free radical levels is the modulation of mitochondrial morphology. Under pathologic stress the mitochondrial population shifts to a highly fragmented state, generating excessively small organelles that lack essential metabolites and proteins required for normal function [81,91,92]. There is an activation of mitochondrial fission in the early reperfusion period that is followed by additional increased fragmentation of neuronal mitochondria [91,92]. This excessive mitochondrial fragmentation was observed particularly in the hippocampal CA1 neurons that represent the cells vulnerable to global ischemic insult [91]. Interestingly, mitochondria in the hippocampal CA3 and DG subregions were more resistant to ischemia and were able to refuse and regain their preischemic morphology after the first initial fragmentation event [91]. When animals were treated with NAD^+^ precursor NMN, the excessive mitochondrial fragmentation was prevented in cells within all hippocampal subregions [56], and there was dramatic neuroprotection observed in mice [65]. Furthermore, NMN administration also reduced ROS accumulation in genetic mouse models of heart failure [93] and myocardial ischemia/reperfusion injury [94], following hypoglycemia-induced brain damage [95], and in a brain focal ischemia model [96].

All these data suggest that the increase in ROS generation during late recovery period following ischemia can be reversed by normalizing the mitochondria and cellular NAD^+^ metabolism.

## 5. Therapeutic Approaches to Reduce Mitochondrially-Generated ROS

A significant fraction of the research in field of mitochondrial free radical metabolism was focused primarily on the mitochondrial respiratory chain complex-dependent increase in ROS production due to stress-induced defects. Therefore, a number of the therapeutic approaches were proposed to scavenge the intramitochondrial free radicals and eliminate their adverse effects by delivering mitochondria-targeted antioxidant molecules. Although mitochondria-targeted antioxidants showed promising effects [97,98,99,100], there are drawbacks in this approach [101]. The accumulation of the mitochondria-targeted compounds is driven by the mitochondrial membrane potential. Mitochondria with higher membrane potentials (more negative) will take up more antioxidants. However, under pathological conditions the dysfunctional mitochondria, due to damage to the respiratory complexes, likely produce the most of ROS. These mitochondria will carry a lower membrane potential, and therefore a higher dose of antioxidants needs to be administered to achieve effective suppression of ROS overproduction. However, this will lead to the overloading of functionally non-compromised, healthy mitochondria. Thus, the level of ROS in healthy mitochondria will be reduced below the physiological levels, causing adverse effects on downstream signaling pathways [101].

Therefore, application of antioxidant compounds that will target mitochondria independently of membrane potential or administration of compounds that are able to reverse the primary cause of increased ROS levels may offer a safer approach for treatment.

## 6. Conclusions

It is commonly accepted that increased mitochondrial free radical generation can significantly contribute to pathophysiology triggered by acute or neurodegenerative brain injury. Our published data show that the delayed increase in brain tissue ROS levels following an acute brain injury is due to inhibition of the mitochondrial antioxidant mechanisms triggered by mitochondrial NAD^+^ catabolism and consequent hyperacetylation of enzymes, whose activities are essential for superoxide detoxification. Administration of NMN prevents the depletion of mitochondrial NAD^+^ pools and its downstream negative effects. This represents a new therapeutic approach that reduces the excessive generation of ROS by mitochondria, inhibits the pathologic mitochondrial fragmentation, and dramatically improves the outcome following global cerebral ischemia.

## Figures and Tables

**Figure 1 brainsci-10-00449-f001:**
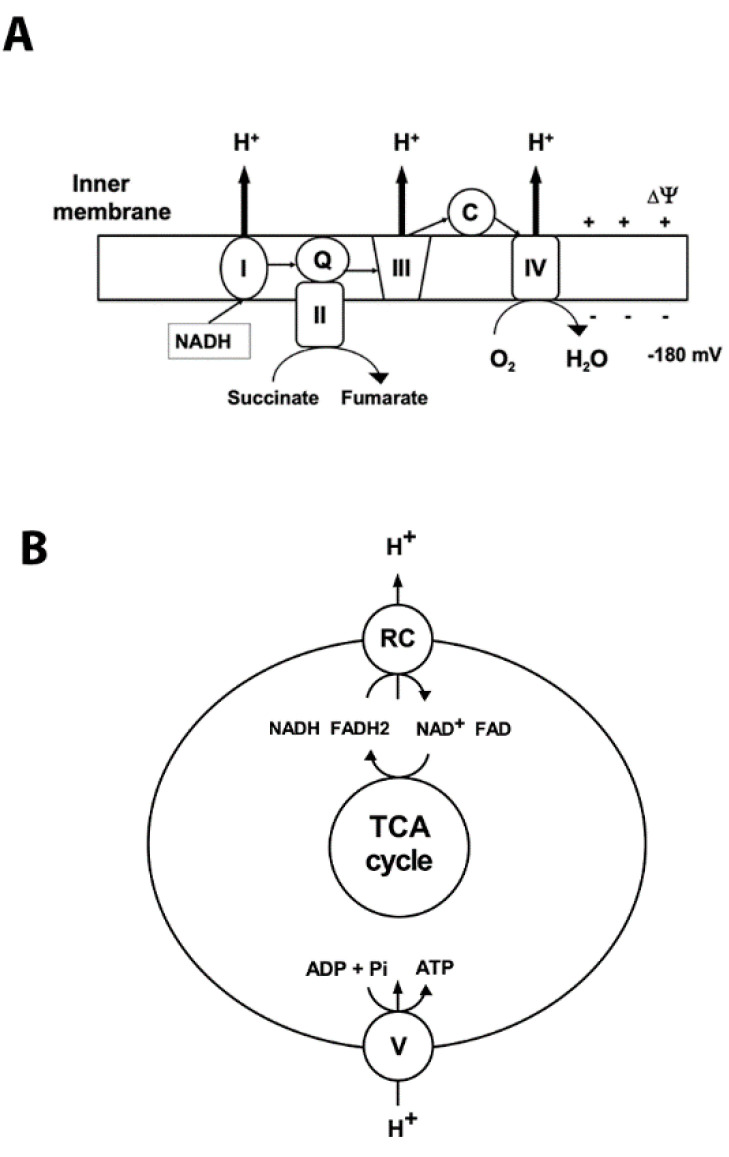
Mitochondrial oxidative phosphorylation. Oxidative phosphorylation generates ATP using the electrochemical potential (comprised of chemical gradient of hydrogen ions and the membrane potential ∆03A8) formed by the respiratory chain complexes in the inner mitochondrial membrane. (**A**) Complex I (NADH: ubiquinone oxidoreductase (I)) accepts electrons from NADH, and together with complex II (succinate: ubiquinone oxidoreductase (II)), which oxidizes succinate to fumarate, transfers electrons to ubiquinone (Q). Then the electrons are accepted by complex III (III) and transferred to complex IV (IV) via cytochrome C (C). At complex IV oxygen is reduced to water. Respiratory chain (RC) complexes I, III, and IV pump hydrogen ions (H^+^) across the inner membrane to generate the protomotive force that drives complex V to synthetize ATP. The thin arrows represent the transfer of the electrons; bold arrows show the hydrogen ions’ transport across the inner membrane. (**B**) Schematic illustration of coupling between activity of respiratory complexes (RC) and ATP production by complex V (ATP synthase).

**Figure 2 brainsci-10-00449-f002:**
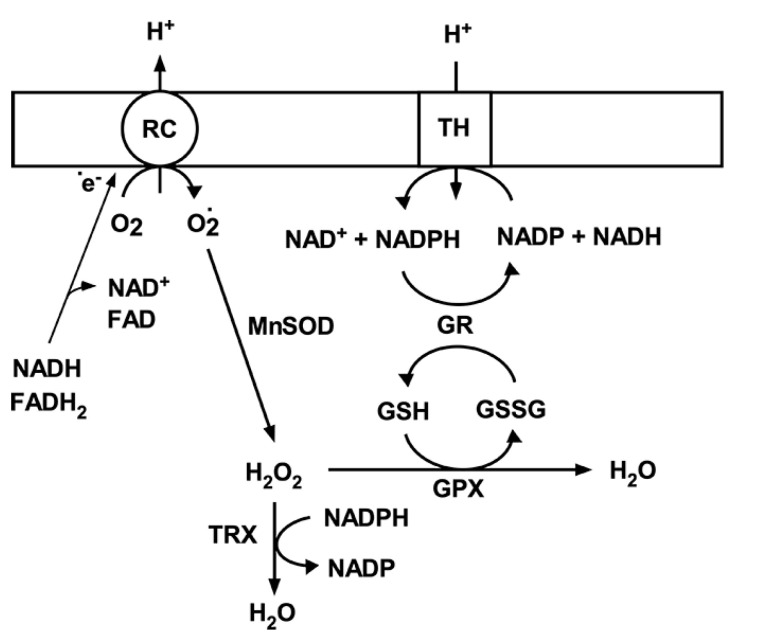
Mitochondrial antioxidant system. Superoxide (O_2_^•^^-^) generated by respiratory chain complexes (RC) is dismutated to hydrogen peroxide (H_2_O_2_) by MnSOD. H_2_O_2_ is then converted to water either by glutathione peroxidase using reduced glutathione (GSH) or thioredoxin peroxidase (TRX). Oxidized glutathione (GSSG) is then reduced back to GSH by glutathione peroxidase (GPX) using NADPH as an electron donor. NADP is then reduced by transhydrogenase; that is driven by mitochondrial membrane potential and transferring electrons from NADH.

**Figure 3 brainsci-10-00449-f003:**
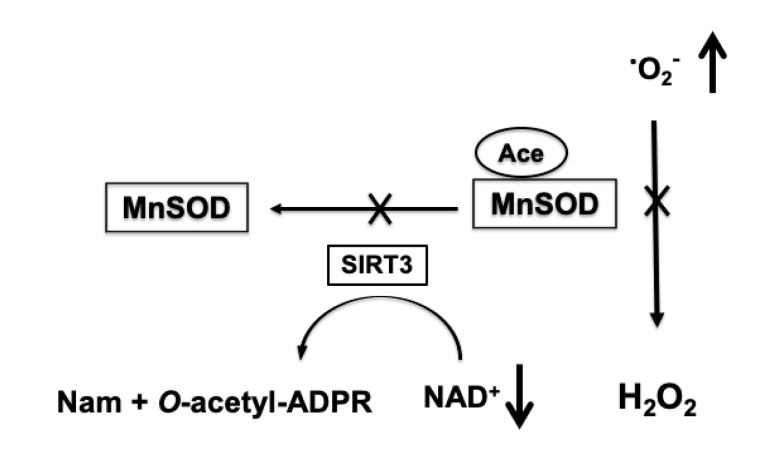
NAD^+^-dependent increase in mitochondrial superoxide production (^•^O_2_^-^). Ischemia-induced depletion of mitochondrial NAD^+^ pools leads to reduced activity of SIRT3 that transfers the acetyl group (Ace) from MnSOD to the ADP ribose (ADPR) moiety of NAD^+^ and forms *O*-acetyl-ADPR and nicotinamide (Nam). The hyperacetylation-induced inhibition of MnSOD results in insufficient removal of superoxide generated by mitochondria causing an increase in ROS levels.

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
