# Peer review of "Role of NAD+—Modulated Mitochondrial Free Radical Generation in Mechanisms of Acute Brain Injury"

_brainsci, 2020, doi:10.3390/brainsci10070449_

Round 1

Reviewer 1 Report

The review article by Klimova et al. titled, "Role of NAD+ - modulated Mitochondrial Free Radical Generation in Mechanisms of Acute Brain 3 Injury" discusses the role NAD depletion following IR injury leads to mitochondrial dysfunction. The review briefly discusses the sources and sinks of mitochondrial oxygen free radicals before going into NAD depletion and protein acetylation that occurs in brain IR injury. The review is relatively easy to read but needs improvement before I can endorse it for publication. One suggestion is to expand on the complex I and NMN relevant portions of the review, as all other information has been in many prior reviews from other authors. For detailed comments, see below.

Consider changing FADH2 and FAD to UQH2 and UQ, as these are the mobile electron carrier analogues to NADH and cyt c.

line 71 "During this process NAD+ and FAD are released." FAD is not released from CII. It's covalently bound. Fumarate is released.

line 81 Dihydrolipoamide dehydrogenase is the common ROS source in aKGDH, PDH, and BCKDH.

line 87 SDH may be a significant producer of ROS during pathological conditions. More recent studies than those cited by the authors discuss the importance of SDH. For details, see the works by Brand, Vinogradov, and Drose. These newer findings should be folded into the review and discussed.

line 112 IDH2 and ME (NADP-malic enzyme) also produce NADPH in mitochondria. Which NADPH generating pathway is the dominant one in brain? Or do they all share equal contributions to NADPH generation?

line 129 This section needs some reorganization. The two phases not clearly laid out.

line 154 change "(for review see [29]." to "(for review, see [29])."

line 160 change "(for review see [29])." to "(for review, see [29])."

line 160 change "catalase" to "catalyze".

lines 160-164, are there other, independent studies the authors can cite to support the claim that FMN-deficient complex I is the direct cause of the increased brain tissue oxygen tension? If not, then this should not be stated so matter-of-factly.

line 165 this paragraph needs better organization, it covers too many distinct topics, e.g. DNA damage and NAD+ depletion. I suggest splitting into two paragraphs.

line 247, section 8. Therapeutic approaches to reduce mitochondrially generated ROS. I suggest refocusing this to include more therapeutic approaches used to treat IR induced mitochodnrial dysfunction and not focus soely on antioxidant therapies. There's wealth of data from nearly two dozen studies on NMN therapy, some of which should be included here. I found two instances describing NMN results and they are only from the authors' own studies.

line 256 The claim that most ROS comes from mitos with lower membrane potential needs more explication.

Author Response

Reviewer 1:

  1. Consider changing FADH2 and FAD to UQH2 and UQ, as these are the mobile electron carrier analogues to NADH and Cyt C.

We have rewritten the first part of section 1, and also modified the figure 1.

  1. Line 71 "During this process NAD+ and FAD are released." FAD is not released from CII. It's covalently bound. Fumarate is released.

As mentioned above this section was rewritten (Line 7-80).

  1. Line 81 Dihydrolipoamide dehydrogenase is the common ROS source in aKGDH, PDH, and BCKDH.

We address this comment by adding the following sentence: “It should be noted that dihydrolipoamide dehydrogenase (DLD) as a component of PDH, aKGDH, and branched-chain αketoacid dehydrogenase complex is a common source of ROS.” (Line 87-89)

  1. Line 87 SDH may be a significant producer of ROS during pathological conditions. More recent studies than those cited by the authors discuss the importance of SDH. For details, see the works by Brand, Vinogradov, and Drose. These newer findings should be folded into the review and discussed.

We expanded the discussion regarding the contribution of Complex II to ROS generation as follows: “Furthermore, under specific conditions of low succinate concentration and a downstream block of the electron flow in the respiratory chain, Complex II can also produce high levels of ROS [16], [17], [18].”  (Line 96-98)

  1. Line 112 IDH2 and ME (NADP-malic enzyme) also produce NADPH in mitochondria. Which NADPH generating pathway is the dominant one in brain? Or do they all share equal contributions to NADPH generation?

We stated in the manuscript that NADP-malic enzyme and NADP-dependent isocitrate dehydrogenase (IDH2) also converts NADP+ to NADPH. (Line 152-153) 

  1. Line 129 This section needs some reorganization. The two phases not clearly laid out.

We have modified this section. (Line 182-186)

  1. Line 154 change "(for review see [29]." to "(for review, see [29])."

Corrected.

8.Line 160 change "(for review see [29])." to "(for review, see [29])."

Corrected.

  1. Line 160 change "catalase" to "catalyze".

Corrected to catalyze. (Line 204)

  1. Lines 160-164, are there other, independent studies the authors can cite to support the claim that FMN-deficient complex I is the direct cause of the increased brain tissue oxygen tension? If not, then this should not be stated so matter-of-factly.

We have re-written this part as follows: “Generally, the inhibition of the mitochondrial respiration will reduce consumption the oxygen delivered to the post-ischemic tissue, resulting in an increased brain tissue oxygen tension. This rise in brain tissue oxygen levels during reperfusion was observed following transient focal ischemia with up to 60 % increase above the physiological values [71]. These hyperoxic conditions then further facilitate the high ROS generation.”  (Line 234-238)

  1. Line 165 this paragraph needs better organization, it covers too many distinct topics, e.g. DNA damage and NAD+ depletion. I suggest splitting into two paragraphs.

We have divided the paragraph into two sections. (Line 258)

  1. Line 247, section 8. Therapeutic approaches to reduce mitochondrially generated ROS. I suggest refocusing this to include more therapeutic approaches used to treat IR induced mitochodnrial dysfunction and not focus solely on antioxidant therapies. There's wealth of data from nearly two dozen studies on NMN therapy, some of which should be included here. I found two instances describing NMN results and they are only from the authors' own studies.

We have reformulated the therapeutic approach section and updated the references that are linked to NMN effect. (Line 308-311, Line 315-324)

  1. Line 256 The claim that most ROS comes from mitos with lower membrane potential needs more explication.

We clarified the underlying conditions for ROS generation by depolarized mitochondria. (Line 323)

Reviewer 2 Report

The manuscript entitled “Role of NAD+ - modulated mitochondrial free radical generation in

mechanisms of acute brain injury” by Klimova et al., is a mini-review in which past and recent data from the author’s lab and other researcher groups on the role of mitochondria-generated ROS  in acute cerebral injury are thoroughly analyzed.

In the reviewer’s opinion, this is a concise but comprehensive review of the mechanism of ROS generation and metabolism in mitochondria during post-ischemic reperfusion. The scope of the present manuscript fits very well to the audience of Brain Science There is no major criticism, but several minor comments:

  1. 1 and Page 2 Lines 68-71. While it is very common to depict FAD/FADH2 couple as part of SDH functioning, the concept is misleading. Indeed flavin is a natural covalently-bound cofactor of SDH. However, the same stands for KGDHC, complex I, and pyruvate dehydrogenase but those flavins are never shown in similar schemes. The confusion can arise from considering FAD/FADH2 couple as a _substrate_, the same way it is shown for free NAD+/NADH coupler for complex I. For the benefit of the reader, it should be amended.

  1. Abbreviations should be explained at the first use (for example Page 2, Line 68 - FADH2)

  1. Fig 1. There is another Q-reducing enzyme present in brain mitochondria, namely glycerol 3-phosphate dehydrogenase. In preparations of brain mitochondria, this enzyme also supplies electrons to the ubiquinone pool and it can significantly contribute to mtROS production per se and also support ROS-generating RET in complex I.  For the benefit of the reader, this could be discussed and relevant references can be added to the discussion:  PMID: 3711892,  PMID: 9466828, PMID: 3711892, PMID: 3711892.

  1. Page 3 Line 87. It is generally accepted that Complex III contributes to mitochondrial ROS generation. Historically, this came from the pioneering work of B. Chance (PMID: 3711892) where H2O2 generation was measured in the presence of antimycin and was assigned to complex III. However, in physiological conditions, the contribution of this enzyme is very likely to be negligible unless certain mutations are present. This should be noted.

  1. Page 3 Line 84. While contribution Shc (p66Shc) protein in mitochondrial ROS production was vigorously discussed in the field, there has been very little evidence of the direct involvement of this protein in a redox reaction with oxygen. It does not exclude its _prooxidant_ activity, but it cannot be discussed among direct ROS _generators_, such as redox-active enzymes.

  1. When referring to the effects of ischemia (sections 4-5 and 7) possible decrease of adenylate and NAD(P) pools due to enzymatic degradation may be described in terms of the effect on postischemic mitochondrial metabolism.

  1. Page 6 Lines 224-228. These two sentences should be rephrased. In the present form, they make the impression that mito-eYFP mice were injected with DHE, which is not the case.

Author Response

Reviewer 2:

  1. Page 1 and Page 2 Lines 68-71. While it is very common to depict FAD/FADH2 couple as part of SDH functioning, the concept is misleading. Indeed flavin is a natural covalently-bound cofactor of SDH. However, the same stands for KGDHC, complex I, and pyruvate dehydrogenase but those flavins are never shown in similar schemes. The confusion can arise from considering FAD/FADH2 couple as a _substrate_, the same way it is shown for free NAD+/NADH coupler for complex I. For the benefit of the reader, it should be amended.

 We have rewritten the first part of section 1 and also modified the figure 1.

  1. Abbreviations should be explained at the first use (for example Page 2, Line 68 - FADH2)

We have defined the abbreviation throughout the manuscript at the first use.

  1. Fig 1. There is another Q-reducing enzyme present in brain mitochondria, namely glycerol 3-phosphate dehydrogenase. In preparations of brain mitochondria, this enzyme also supplies electrons to the ubiquinone pool and it can significantly contribute to mtROS production per se and also support ROS-generating RET in complex I.  For the benefit of the reader, this could be discussed and relevant references can be added to the discussion:  PMID: 3711892,  PMID: 9466828, PMID: 3711892, PMID: 3711892.

We did mention the glycerol-3-phosphate dehydrogenase (page 3, line 86). Now we extended the list of references as the reviewer suggested.

  1. Page 3 Line 87. It is generally accepted that Complex III contributes to mitochondrial ROS generation. Historically, this came from the pioneering work of B. Chance (PMID: 3711892) where H2O2 generation was measured in the presence of antimycin and was assigned to complex III. However, in physiological conditions, the contribution of this enzyme is very likely to be negligible unless certain mutations are present. This should be noted.

We address the complex III ROS generation conditions by stating: However, significant contribution of Complex III was observed mainly under artificial conditions when using Complex III inhibitor Antimycin A [6]. (Line 95-96)

  1. Page 3 Line 84. While contribution Shc (p66Shc) protein in mitochondrial ROS production was vigorously discussed in the field, there has been very little evidence of the direct involvement of this protein in a redox reaction with oxygen. It does not exclude its _prooxidant_ activity, but it cannot be discussed among direct ROS _generators_, such as redox-active enzymes.

We removed the Shc (p66Shc) protein from the sentence. (Line 89)

  1. When referring to the effects of ischemia (sections 4-5 and 7) possible decrease of adenylate and NAD(P) pools due to enzymatic degradation may be described in terms of the effect on postischemic mitochondrial metabolism.

We renumbered the section 5 to 7 as 4.1., 4.2, and 4.3.

  1. Page 6 Lines 224-228. These two sentences should be rephrased. In the present form, they make the impression that mito-eYFP mice were injected with DHE, which is not the case.

This is probably an oversite by the reviewer since the mito-eYFP transgenic animals were injected with DHE (see Klimova et al. Exp Neurology 2020; vol. 325, p. 113144 , PMID 30801823).

Reviewer 3 Report

This is a nice, short review paper by Klimova, Fearnow, and Kristian, regarding the mechanisms of mitochondrial ROS production and detoxification in brain.  This is a very important topic in the field of neurodegenerations and traumatic brain injury. The authors in very concise manner touched the major issues of ROS production and ROS elimination in brain mitochondria. The authors are experts in this area and their view on ROS production/detoxification in mitochondria is very interesting and informative. The paper is very well organized, clearly written, and easy to read. There are just a few minor concerns.  

In Introduction, line 31: Mitochondria are plural form of mitochondrion.  Please, correct, either "Mitochondria are essential organelles..." or "Mitochondrion is an essential organelle..."

Line 50: The electrochemical potential is usually expressed as deltaMu, deltaPsay, used by the authors, is an electrical component (membrane potential) of the electrochemical potential.

Line 89:  The authors are focused on superoxide anion generation and its detoxification mechanisms. However, the authors also mentioned a few other reactive species. It would be helpful if the authors would add a sentence justifying their choice of superoxide anion as a focus of their paper. Probably, this is because superoxide anion is the major form of ROS produced in mitochondria.

Line 152: "...that suggest..." - should be "...that suggests..."

Line 156:  It is very well established that the reverse electron transfer in the electron transport chain is highly dependent on mitochondrial membrane potential and slows significantly with a very mild depolarization, which is most likely present at the early stage of reperfusion due to, for example, opening of the PTP (see text of the manuscript below). How would the authors reconcile this discrepancy?

Author Response

Reviewer 3:

  1. In Introduction, line 31: Mitochondria are plural form of mitochondrion.  Please, correct, either "Mitochondria are essential organelles..." or "Mitochondrion is an essential organelle..."

 We corrected the sentence, changed from organelle to organelles.

  1. Line 50: The electrochemical potential is usually expressed as deltaMu, deltaPsay, used by the authors, is an electrical component (membrane potential) of the electrochemical potential.

We defined the electrochemical potential as the chemical gradient of hydrogen ions and the membrane potential DY. (Figure 1 legend)

  1. Line 89:  The authors are focused on superoxide anion generation and its detoxification mechanisms. However, the authors also mentioned a few other reactive species. It would be helpful if the authors would add a sentence justifying their choice of superoxide anion as a focus of their paper. Probably, this is because superoxide anion is the major form of ROS produced in mitochondria.

We added the following sentence: “Since the superoxide anions are the major form of ROS generated by mitochondria, we will focus mainly on its generation and detoxification mechanisms.” (Line 100-101)

  1. Line 152: "...that suggest..." - should be "...that suggests..."

Corrected as suggested. (Line 196)

  1. Line 156: It is very well established that the reverse electron transfer in the electron transport chain is highly dependent on mitochondrial membrane potential and slows significantly with a very mild depolarization, which is most likely present at the early stage of reperfusion due to, for example, opening of the PTP (see text of the manuscript below). How would the authors reconcile this discrepancy?

We have addressed this comment by the following paragraph: “The opening of the MPT leads to mitochondrial depolarization. Since the reverse electron transport is reduced under low membrane potential one would expect that the ROS generation by this process will be inhibited. However, our published data demonstrate that brain mitochondria display a clear-cut heterogeneity in sensitivity to MPT [68]. Therefore, it is feasible that a subpopulation of mitochondria that are less sensitive to MPT will contribute to rise in ROS due to reverse electron transport during the early recovery.” (Line 229-233)

Reviewer 4 Report

This is a comprehensive review of the role of mitochondrial ROS production following brain injury. The authors have not only described currently established mechanisms, but have included lesser known pathways that have significant contribution in the understanding of mitochondrial dysfunction. The figures are concise and make difficult pathways easy to interpret for the reader. The only minor corrections needed are for the authors to carefully review this manuscript for inappropriate verb tense and incorrect noun pluralization.

Author Response

Reviewer 4:

  1. The only minor corrections needed are for the authors to carefully review this manuscript for inappropriate verb tense and incorrect noun pluralization.

We have carefully checked the manuscript for English grammar and corrected the mistakes.

Round 2

Reviewer 1 Report

The authors have addressed my comments.